# Fitness Shifts the Balance of BDNF and IL-6 from Inflammation to Repair among People with Progressive Multiple Sclerosis

**DOI:** 10.3390/biom11040504

**Published:** 2021-03-26

**Authors:** Augustine Joshua Devasahayam, Liam Patrick Kelly, John Bradley Williams, Craig Stephen Moore, Michelle Ploughman

**Affiliations:** 1L.A. Miller Centre, Recovery and Performance Laboratory, Faculty of Medicine, Memorial University of Newfoundland, St. John’s, NL A1K 5A1, Canada; augustine.joshua@mun.ca (A.J.D.); lpkelly@mun.ca (L.P.K.); 2Division of BioMedical Sciences, Faculty of Medicine, Memorial University of Newfoundland, St. John’s, NL A1B 3V6, Canada; bwilliams@mun.ca (J.B.W.); craig.moore@mun.ca (C.S.M.)

**Keywords:** multiple sclerosis, rehabilitation, fitness, neurodegeneration, inflammation, biomarkers, neurotrophins, brain-derived neurotrophic factor, cytokines, interleukin-6

## Abstract

Physical sedentarism is linked to elevated levels of circulating cytokines, whereas exercise upregulates growth-promoting proteins such as brain-derived neurotrophic factor (BDNF). The shift towards a ‘repair’ phenotype could protect against neurodegeneration, especially in diseases such as multiple sclerosis (MS). We investigated whether having higher fitness or participating in an acute bout of maximal exercise would shift the balance of BDNF and interleukin-6 (IL-6) in serum samples of people with progressive MS (*n* = 14), compared to matched controls (*n* = 8). Participants performed a maximal graded exercise test on a recumbent stepper, and blood samples were collected at rest and after the test. We assessed walking speed, fatigue, and maximal oxygen consumption ($\overset{\cdot }{V}O_{2\max}$). People with MS achieved about 50% lower $\overset{\cdot }{V}O_{2\max}$ (*p* = 0.003) than controls. At rest, there were no differences in BDNF between MS and controls; however, IL-6 was significantly higher in MS. Higher $\overset{\cdot }{V}O_{2\max}$ was associated with a shift in BDNF/IL-6 ratio from inflammation to repair (R = 0.7, *p* = 0.001) when considering both groups together. In the MS group, greater ability to upregulate BDNF was associated with faster walking speed and lower vitality. We present evidence that higher fitness indicates a shift in the balance of blood biomarkers towards a repair phenotype in progressive MS.

## 1. Introduction

Globally, more than 2.3 million people live with multiple sclerosis (MS), and of those, over one million people have a progressive form of MS, a type with fewer inflammatory relapses but more prominent neurodegeneration [1]. Why some people remain stable and others progress is not clear, and experts in the field have prioritised research to accelerate the development of effective therapies for people with progressive MS [2,3,4]. One of these efforts is to create novel rehabilitation strategies that combine the reparative, neuroplastic, cardiorespiratory, and metabolic benefits of aerobic exercise in order to promote brain repair [5].

Physical sedentarism is linked to elevated levels of circulating inflammatory markers, such as interleukin-6 (IL-6), a cytokine that increases brain inflammation [6]. A recent study reported that heightened levels of IL-6 in cerebrospinal fluid were associated with blunted capacity for neuroplasticity in 150 people with MS [7]. These inflammatory cytokines are toxic to the brain [8] but can be inhibited by participation in physical exercise [9]. While IL-6 is pro-inflammatory and linked to neurodegeneration, the neurotrophin brain-derived neurotrophic factor (BDNF), produced by both glial cells and contracting muscle, regulates synaptic change and use-dependent brain plasticity [10]. Aerobic exercise, by upregulating neurotrophins [11,12,13] and altering cytokine levels [14,15,16,17], could be neuroprotective in MS, thereby facilitating neuroplasticity and protecting against functional decline [11,13,18].

In people with MS, studies have demonstrated a dose-response relationship on serum concentrations of neurotrophin BDNF [11,12,13] and cytokine IL-6 [11] after a graded exercise test (GXT), supporting that aerobic training interventions could have direct effects on the neuro-immune axis [11,18]. For example, combined aerobic and Pilates training increased resting serum BDNF levels, while simultaneously improving walking endurance, balance, and fatigue in people with relapsing-remitting MS [19]. A systematic review reported that among people with MS, aerobic training significantly altered peripheral levels of cytokines IL-6, IL-10, interferon-γ, and tumor necrosis factor (TNF) [20]. Whether neurotrophins and cytokines remain responsive to exercise among people with greater disability and with a more progressive form of MS is not clear [20,21]. Despite established reciprocal actions of BDNF and IL-6 to promote neuronal survival [22], representing inflammation and neurotrophic markers as a ratio, for example, BDNF/IL-6 ratio, has never been reported in MS. A recent study of 1833 older people with diabetes suggested that a shifting of the ratio between BDNF and dipeptidyl peptidase-4 (inflammation) could be an important biomarker of cognitive impairment [23]. Analysing the relationships between potential biomarkers such as resting and exercise-induced levels of neurotrophin (BDNF) and cytokine (IL-6) and functional measures (walking speed [24,25,26,27], fatigue [28,29], and aerobic fitness (maximal oxygen consumption ($\overset{\cdot }{V}O_{2}$) [30,31,32]) may help determine whether the neurotrophin BDNF and cytokine IL-6 are appropriate rehabilitative targets and surrogate markers of recovery and neuroplasticity in progressive MS [33,34,35,36].

As a first step, we investigated to what extent GXT would change the circulating levels of BDNF, IL-6, and their ratio, in a group of people with similar levels of MS-related walking disability, compared to matched controls. Next, we examined whether these resting and exercise-induced serum blood markers were associated with important outcomes, i.e., walking speed, fatigue, and aerobic fitness (maximal $\overset{\cdot }{V}O_{2}$ and heart rate).

## 2. Materials and Methods

### 2.1. Participants

The study was approved by the Health Research Ethics Board (HREB #20162300). Following informed written consent, we recruited patients who attended outpatient physiotherapy or the MS clinic. Patients were eligible if they (1) had a confirmed diagnosis of progressive MS by a neurologist using McDonald criteria [37], (2) had an expanded disability status scale (EDSS) score of 6 or 6.5 (used bilateral walking aids), (3) were stable without any relapse or other health event in the previous 90 days, (4) did not have comorbid cerebrovascular and lung conditions, and (5) were not receiving glucocorticoids. Healthy controls were matched for sex and age (±3 years) and recruitment of controls ceased once all the MS participants were matched. All participants completed the physical activity readiness questionnaire (PAR-Q) to ensure safety during exercise [38,39]. Those participants who failed PAR-Q were referred to a physician for a physical activity readiness medical examination (PARmed-X) [40]. With the alpha set at 5% and a power of 80%, the minimum sample size was estimated to be between 13 and 37 to detect the effects of exercise-induced serum BDNF in people with MS [11,13].

### 2.2. Performance Outcomes

We measured comfortable walking speed by asking participants to walk overground on a 15-feet long path for 5 min. The distance was divided by the time to complete and represented as m·s^−1^. Self-reported fatigue was measured using the vitality/energy/fatigue sub-scale of the 36-item short-form health survey (SF-36) [41,42]. The vitality/energy/fatigue sub-scale of SF-36 rates feelings of energy/fatigue as a unidimensional construct on a continuum capturing both negative (fatigue) and positive (energy) states [43]. Values were weighted/transformed according to published procedures to obtain a score ranging from 0 to 100 with lower scores indicating worse fatigue and higher scores indicating greater energy levels [43].

All participants were assessed to determine their maximal $\overset{\cdot }{V}O_{2}$ and heart rate during GXT. The participants were advised not to consume food for at least four hours preceding the GXT. We used a total body recumbent stepper as per protocol adapted by Kelly et al. [44], wearing a face mask connected via tubing to a breath-by-breath metabolic cart (Moxus Metabolic Systems, AEI Technologies, Inc., Pittsburgh, PA, USA). Participants were instructed to maintain 80 steps per minute during GXT and the workload was increased in ~20-watt increments every 2 min, starting from load level 3 (21 watts) until exhaustion [44]. They were considered to have attained maximal $\overset{\cdot }{V}O_{2}$ if at least two of the following criteria were met: (1) $\overset{\cdot }{V}O_{2}$ plateau (failure to increase $\overset{\cdot }{V}O_{2}$ by 150 mL·min^−1^) [45] with increasing workload (inability to maintain workload/stepping frequency of 80 per minute) [44], (2) respiratory exchange ratio > 1.10 [45], (3) >90% age-predicted maximal heart rate [45], and (4) >8.0 modified Borg’s rating of perceived exertion [45].

### 2.3. Blood Samples

Blood samples were drawn from the median cubital vein immediately before and following GXT in two 5 mL serum vacutainers [46]. Samples were left to clot for 30–60 min, centrifuged at 2200 g for 10 min, and the collected serum was stored frozen at −80 °C. Serum levels of BDNF and cytokine IL-6 were measured using enzyme-linked immunosorbent assay sets for human BDNF (R&D Systems Inc. Minneapolis, MN, USA) and IL-6 (BD Biosciences, San Diego, CA, USA) as per the manufacturer’s protocols. Serum levels of TNF were also measured in all samples (BD Biosciences, San Diego, CA, USA), and the levels were below the detectable range.

### 2.4. Data Analysis

The assumptions of normality were checked for all variables by inspecting the distribution visually using histograms and box plots, and through Shapiro-Wilk tests (*p* > 0.01) [47,48]. To examine whether exercise changed BDNF, IL-6, and BDNF/IL-6 ratio, we used repeated-measures ANOVA (2 (Groups: MS and control) × 2 (Time: pre- and post-exercise)) or non-parametric equivalent if assumptions of normality were violated. Effect sizes were expressed as partial eta squared (η^2^), where η^2^ of 0.01 was considered a small effect, 0.06 a moderate effect, and 0.14 a large effect [49].

Group differences at baseline were examined using independent t-tests for continuous variables, after checking the homogeneity of variance using Levene’s tests (*p* < 0.05). If assumptions of normality and equal variances were not met, independent samples Mann-Whitney U tests were used to detect the difference between groups. Pearson χ2 test was used for categorical variables to examine group differences at baseline, whereas if one or more of the cells had an expected frequency of five or less, Fisher exact test was used. The relationships between the potential biomarkers (resting and GXT-induced BDNF, IL-6, and BDNF/IL6 ratio) and the functional measures in MS (comfortable walking speed, fatigue, and maximal $\overset{\cdot }{V}O_{2}$) were analysed using Spearman’s rank correlation coefficient (r_s_).

The minimum detectable concentrations of BDNF and IL-6 in serum were determined to be 0.0234 ng·mL^−1^ and 0.0031 ng·mL^−1^, respectively. The values below the detection limit were replaced by half the lowest concentration recorded for the respective analyte within each group [50,51,52]. The serum concentrations of potential biomarkers were expressed as a ratio to characterise the shift in the balance of blood biomarkers before and after GXT.

## 3. Results

### 3.1. Participant Characteristics

A total of 38 individuals were contacted for this study (22 with definite MS, 16 age/sex-matched controls). Of those, 16 participants were excluded, four subjects with MS who did not use walking aids, one subject with MS who did not wish to complete the exercise, six controls who did not match for age, and five others who were unable to be contacted after their first telephone call (three MS, two controls). We, therefore, recruited 14 people with MS and 8 age/sex-matched controls. One control subject dropped out after enrolment. There were no statistically significant differences between the MS group and controls in terms of age, sex distribution, and body mass index (Table 1). In the MS group, the total number of years lived with a confirmed diagnosis of MS ranged from 3 to 31 years old. All participants but two (1 MS and 1 control subject) passed the PAR-Q [38,39], however, they were included in the study after the completion of PARmed-X [40].

Participants with progressive MS achieved about 50% lower maximal $\overset{\cdot }{V}O_{2}$ (Z = −3.283, *p* = 0.0003) (Table 1). The maximal workload achieved by participants with progressive MS during GXT was about 37% of that achieved by controls (Z = 3.209, *p* = 0.00049). All participants reported performing the test to maximal volitional exhaustion, and there was no significant difference in the rating of perceived exertion measured immediately after GXT between groups (Z = −0.684, *p* = 0.585).

Blood samples were collected at rest and again within 7 min (246.89 ± 137.04 s) of GXT termination. The mean blood collection time after GXT was not significantly different (Z = −0.517, *p* = 0.616) between MS (259.08 ± 113.18 s) and controls (222.5 ± 186.0 s). Serum BDNF levels in both participants with progressive MS and matched controls were within the detectable ranges. IL-6 was detectable in 60.5% of samples tested (<0.0001 ng·mL^−1^).

### 3.2. Resting and Post-Exercise Serum BDNF

In terms of serum BDNF, there was no significant main effect of group (F_(1,17)_ = 0.007, *p* = 0.93, η_p_^2^ = 0.0004), time (F_(1,17)_ = 0.004, *p* = 0.95, η_p_^2^ = 0.0002), or group X time interaction (F_(1,17)_ = 0.002, *p* = 0.97, η_p_^2^ = 0.0001), indicating similar serum BDNF levels before and after GXT in MS and controls (Table 1, Figure 1a,b). Considering MS and controls together, resting serum BDNF did not correlate with walking speed, fatigue, and fitness (*p* values > 0.05) (Figure 2a–c). When considering only the MS group, higher exercise-induced elevation in serum BDNF was significantly related to faster walking speed (r_s_ = 0.62, *p* = 0.043) and lower vitality measured using vitality/energy/fatigue subscale of SF-36 (r_s_ = −0.58, *p* = 0.046), but not fitness (maximal $\overset{\cdot }{V}O_{2}$, r_s_ = 0.45, *p* = 0.138, or maximal heart rate, r_s_ = 0.22, *p* = 0.484) (Figure 3a–d).

### 3.3. Resting and Post-Exercise Serum IL-6

In terms of serum IL-6, there was no significant main effect of group (F_(1,17)_ = 2.45, *p* = 0.14, η_p_^2^ = 0.13) or group X time interaction (F_(1,17)_ = 0.24, *p* = 0.63, η_p_^2^ = 0.014); however, there was a significant main effect of time (F_(1,17)_ = 12.51, *p* = 0.003, η_p_^2^ = 0.42). The participants with MS had significantly higher IL-6 than controls measured at rest (Z = 2.57, *p* = 0.01). GXT elicited a significant elevation in serum levels of IL-6 in MS (Z = 2.83, *p* = 0.005), but not in controls (Z = 1.36, *p* = 0.17: Table 1, Figure 1c,d). Higher resting IL-6 was significantly related to slower walking speed (r_s_ = −0.51; *p* = 0.03), lower energy levels (higher fatigue) (r_s_ = −0.48; *p* = 0.04), and poorer fitness (r_s_ = −0.56; *p* = 0.01) (Figure 2d–f). Exercise-induced IL-6 did not correlate with walking speed, fatigue, and fitness (maximal $\overset{\cdot }{V}O_{2}$ and maximal heart rate) (*p* values > 0.05).

### 3.4. Resting and Post-Exercise Serum BDNF/IL-6 ratio

There was a significant main effect of group on BDNF/IL-6 ratio (F_(1,17)_ = 13.69, *p* = 0.002, η_p_^2^ = 0.45). BDNF/IL-6 ratio scores were significantly lower in MS before GXT (*p* = 0.004), but not after GXT (*p* = 0.12) when compared to controls. Although there was no significant main effect of time on BDNF/IL-6 ratio (F_(1,17)_ = 0.81, *p* = 0.38, η_p_^2^ = 0.045), one bout of GXT resulted in a significant decrease in BDNF/IL-6 ratio in MS (*p* = 0.01), but not in controls (*p* = 0.74, Table 1, Figure 1e,f). Additionally, there was no significant interaction between time and group in terms of BDNF/IL-6 ratio (F_(1,17)_ = 0.52, *p* = 0.48, η_p_^2^ = 0.03). Lastly, higher resting BDNF/IL-6 ratio was significantly related to faster walking speed (r_s_ = 0.57; *p* = 0.013), higher energy levels (lower fatigue) (r_s_ = 0.55; *p* = 0.015), and higher fitness (r_s_ = 0.57; *p* = 0.011) (Figure 2g–i). Exercise-induced changes in BDNF/IL-6 ratio did not correlate with walking speed, fatigue, or fitness (maximal $\overset{\cdot }{V}O_{2}$ and maximal heart rate) (*p* values >0.05).

## 4. Discussion

Recent research supports that exercise could provide neuroprotection in MS by interacting with the neuro-immune axis [53,54,55]. Therefore, the main aim of this study was to compare serum levels of BDNF, IL-6, and their ratio, at rest and after GXT between people with progressive MS and matched controls and determine relationships to walking speed, fatigue, and aerobic fitness. We report four main findings. Firstly, people with progressive MS were severely deconditioned, with fitness levels well below that required to comfortably carry out everyday activities [56,57] (Table 1). Secondly, other than IL-6 which was higher in MS subjects and was further increased with exercise, there were no differences in resting and exercise-induced levels of BDNF, IL-6, and BDNF/IL-6 ratio between groups. Thirdly, considering MS and controls together, we found that higher resting BDNF/IL-6 ratio was significantly related to faster walking speed, lower fatigue, and higher fitness. Lastly, in participants with MS, greater exercise-induced levels of serum BDNF were associated with faster walking speed and higher fatigue.

### 4.1. Aerobic Fitness, Disability, and Expression of Neurotrophins

In our study, all participants performed GXT on a total body recumbent stepper until maximal voluntary exhaustion was achieved (100% of their capacity); yet, we did not detect statistically significant increases in serum BDNF levels. Despite reporting comparable levels of exhaustion at the end of GXT, participants with MS achieved significantly lower maximal workload and peak heart rate. Previous research supports that the release of BDNF in the blood is proportional to the intensity of the exercise [58,59]. It is likely that our MS participants, having extremely low levels of fitness, had blunted capacity to upregulate BDNF. In people with MS with minimal disability (EDSS 2.3 ± 0.2), Gold et al. [13] reported a significant increase in serum BDNF (approximately 1.4 times more) after cycling at a moderate intensity (60% of maximal $\overset{\cdot }{V}O_{2}$) for 30 min. Similarly, Briken et al. [11] reported 1.2 times increase in BDNF in a group of people living with progressive MS with moderate disability (EDSS 4.9 ± 0.8) after 10–20 min of exercise during standardised maximal bicycle ergometer test achieving a peak workload of 97.5 watts. Although our participants with progressive MS achieved 99.7 watts after 7–22 min of exercise on the recumbent stepper, they had, on average, a 0.2% decrease in BDNF after exercise. Lack of BDNF responsiveness to exercise could be related to the fact that our participants had more severe disability (EDSS 6.0–6.5) and lower levels of fitness than that previously reported; they also had 15% lower maximal $\overset{\cdot }{V}O_{2}$ than subjects recruited by Briken et al. (1490.18 mL vs. 1260.9 mL) [11]. However, it is important to note that there were also no increases in serum BDNF in age/sex-matched control subjects who exercised (37.1%) longer than MS participants during GXT, suggesting that the stimulus (GXT) was of insufficient duration (60.4% of Gold et al. [13]) to upregulate BDNF in serum.

### 4.2. Skeletal Muscle and Serum BDNF Induction

Although the brain contributes to almost 75% of the circulating BDNF [58], skeletal muscle is increasingly being recognised as a secretory organ and an important source of BDNF [60,61]. BDNF, in turn, is thought to be transported across the blood-brain barrier to influence brain plasticity [62]. Although we did not measure muscle integrity, our findings support the notion that the ability to upregulate serum BDNF may be related to skeletal muscle. We report that exercise-induced BDNF levels were related to walking speed which is determined by the leg muscle’s ability to propel the body forward [63,64,65,66,67,68,69]. Similarly, among healthy adults who had high fat-free (skeletal muscle) mass, there was greater release and faster recovery of serum BDNF after GXT [70]. Our cohort’s average walking speed was 0.32 m·s^−1^; about one-third of typical gait speed [63] and half that reported among people with MS who walked using a cane [71]. This suggests that, in the slowest walkers (<0.3 m·s^−1^), their muscles were unable to release BDNF. Our findings point to the importance of targeting deconditioning and muscle weakness among people with progressive MS to improve walking and potentially enhance exercise-induced BDNF which could have important benefits on brain health [5].

### 4.3. Factors Influencing Cytokine Responses in MS

We showed that both resting- and exercise-induced levels of IL-6 were greater in MS subjects than in controls. Research suggests that the two measures, resting levels of IL-6 and exercise-induced levels of IL-6, are indicative of entirely different processes. For instance, at rest, B cells derived from people with MS secrete higher than typical levels of IL-6 which appears to contribute to inflammatory-mediated pathogenesis [72]. However, exercise-induced IL-6 may be beneficial. In healthy volunteers, IL-6 released from skeletal muscle with exercise is purported to downregulate TNF, an important mediator of inflammation [73]. Exercise-induced IL-6 is essential for maintaining homeostasis [74]; mediating some of the systemic benefits of exercise [75,76]. For example, in a study examining the acute effects of exercise on IL-6 and macrophages in obese mice, exercise-induced increases in IL-6 were associated with the weakening of M1 phenotype (less inflammatory) in adipose tissue macrophages [77]. Briken et al. [11] reported that following nine weeks of endurance training, people with progressive MS experienced greater elevation (36.2%) in serum IL-6 levels after GXT compared to a wait-list control group (10.3% increase) (*p* = 0.06). In our untrained study group, participants with progressive MS had a 40% increase in serum IL-6 levels after GXT. Taken together, exercise-induced IL-6 seems to have some biological plausibility as a potential rehabilitation biomarker. On the other hand, previous research has shown that serum cytokines in humans are influenced by many lifestyle and behavioural factors, including stress [78], gut microbiome [79], diet [80,81], sleep quality [82,83], diurnal variation [84,85,86], smoking [87], alcohol [88], etc. Despite the variability in IL-6, it appeared that both resting- and exercise-induced levels were responsive to perturbation. Future research should examine whether these levels change longitudinally and whether they align with progression or improvement in MS symptoms.

### 4.4. Pattern of Serum BDNF/IL-6 Ratio

We found that a higher BDNF/IL-6 ratio was associated with faster walking speed, lower fatigue, and higher fitness when considering MS and controls together. We propose that the pattern of resting serum levels of BDNF and IL-6 represent the effort of the immune system to induce repair and restore normalcy. Findings from previous research showed that an imbalance between pro- and anti-inflammatory cytokines demonstrated through a higher ratio of TNF-α/IL-10 and IL-6/IL-10 in the brain of animals exposed to chronic mild stress, may contribute to mood disorders [89]. A study examining 66 patients undergoing hematopoietic stem cell transplantation showed that those with depression had a higher ratio of IL-6/IL-10 than controls [90]. More studies are needed to investigate the role of the ‘inflammation to repair balance’ in individuals who are living with neuroinflammatory conditions such as MS.

## 5. Limitations

Our study has several limitations that must be acknowledged. Firstly, we did not collect information about diet or physical activity patterns to examine whether these factors influenced the measurement of blood biomarkers and $\overset{\cdot }{V}O_{2}$. Secondly, there was substantial variability in resting and exercise-induced levels of BDNF and IL-6. Some of this variability could be related to the fact that the exercise stimulus was of different intensities and durations depending on the person’s level of fitness. We also had a small number of controls. Even though all persons with MS were matched, future research should increase sample sizes in order to account for this variability. Furthermore, we noted an inverse relationship between exercise-induced BDNF and vitality in MS. This counterintuitive finding suggests that future research should examine these blood levels acutely after a standardised exercise session and then longitudinally as a result of long-term training. Lastly, IL-6 is a pleiotropic cytokine that takes part in a wide range of biological activities including inflammation, immune regulation, metabolism, hematopoiesis, and oncogenesis [91]. Considering the fact that IL-6 can take part in multiple, potentially overlapping signaling mechanisms, the interpretation of data related to IL-6 from our study must be limited to the specific context of this research.

## 6. Conclusions

People with progressive MS using walking aids achieved about 50% lower $\overset{\cdot }{V}O_{2\max}$ than controls. At rest, there were no significant differences in BDNF between MS and controls, however, IL-6 was significantly higher in MS. In the MS group, greater ability to upregulate BDNF during maximal exercise was strongly associated with faster walking speed and higher fatigue. Higher $\overset{\cdot }{V}O_{2\max}$ was strongly associated with a shift in BDNF/IL-6 ratio from inflammation to repair when considering both groups together. We present evidence that fitness and exercise indicate a shift in the balance of blood biomarkers towards a repair phenotype even among people who have accumulated significant MS-related disability. How exercise-induced BDNF may influence the neuro-immune axis and interact with the blood-brain barrier is an important area of future research.

## Figures and Tables

**Figure 1 biomolecules-11-00504-f001:**
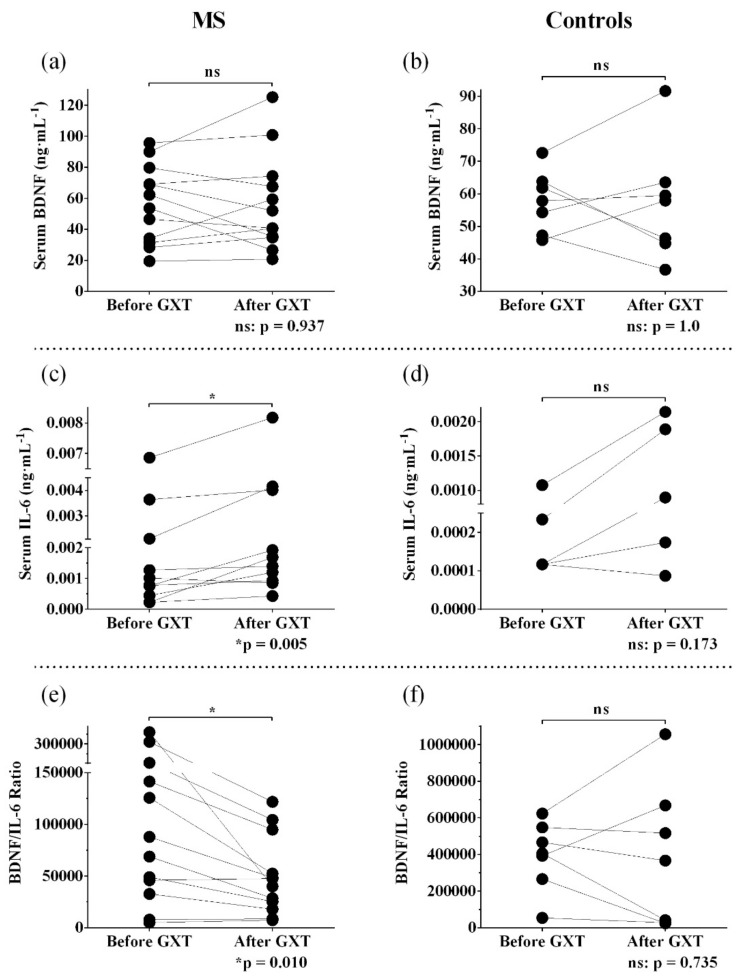
Blood marker responses to graded exercise test. Data presented as individual values. (**a**,**b**): Serum levels of brain-derived neurotrophic factor (BDNF) (ng·mL^−1^) in MS and controls; (**c**,**d**): serum levels of interleukin-6 (IL-6) (ng·mL^−1^) in MS and controls; (**e**,**f**): BDNF/IL-6 ratios in MS and controls; *p* values are from related-samples Wilcoxon signed-rank tests.

**Figure 2 biomolecules-11-00504-f002:**
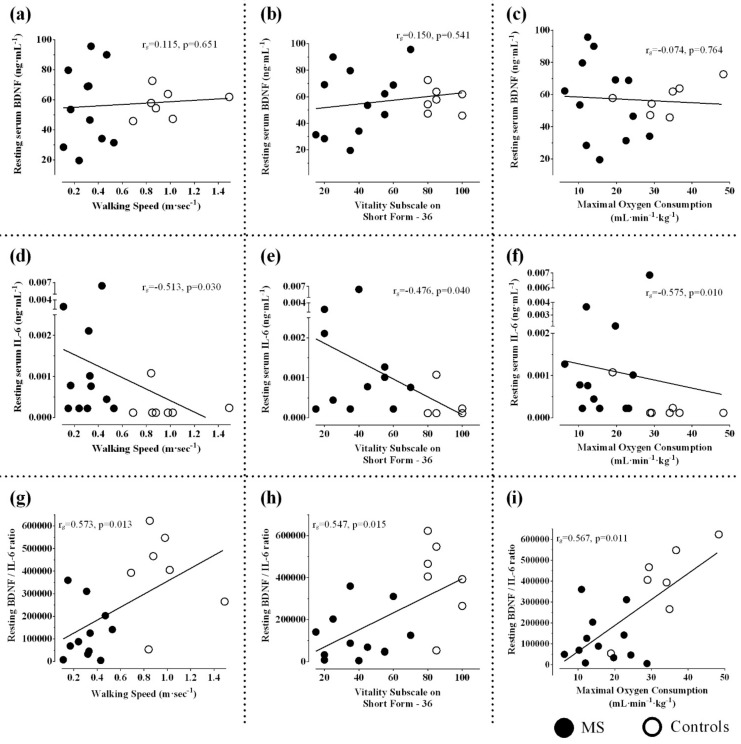
Relationships between biomarkers and functional measures when considering MS and controls together. Data presented as individual values. (**a**) Relationship between self-selected walking speed and resting BDNF; (**b**) relationship between vitality subscale on short form–36 and resting BDNF; (**c**) relationship between maximal oxygen consumption ($\overset{\cdot }{V}O_{2}$) during graded exercise test (GXT) and resting BDNF; (**d**) relationship between self-selected walking speed and resting IL-6; (**e**) relationship between vitality subscale on short form–36 and resting IL-6; (**f**) relationship between maximal $\overset{\cdot }{V}O_{2}$ during GXT and resting IL-6; (**g**) relationship between self-selected walking speed and resting BDNF/IL-6 ratio; (**h**) relationship between vitality subscale on short form–36 and resting BDNF/IL-6 ratio; (**i**) relationship between maximal $\overset{\cdot }{V}O_{2}$ during GXT and resting BDNF/IL-6 ratio; *r* and *p* values are from Spearman’s rank correlation coefficient.

**Figure 3 biomolecules-11-00504-f003:**
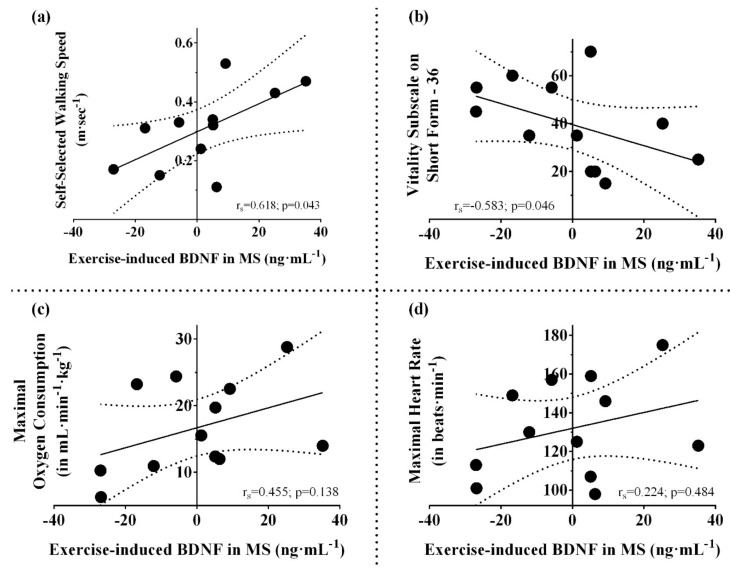
Relationships between exercise-induced BDNF (in ng·mL^−1^) and functional measures in participants with MS. Data presented as individual values. (**a**) Relationship between self-selected walking speed and exercise-induced BDNF; (**b**) relationship between vitality subscale on short form–36 and exercise-induced BDNF; (**c**) relationship between maximal $\overset{\cdot }{V}O_{2}$ during GXT and exercise-induced BDNF; (**d**) relationship between maximal heart rate during GXT and exercise-induced BDNF; the *r* and *p* values are from Spearman’s rank correlation coefficient.

**Table 1 biomolecules-11-00504-t001:** Participant characteristics.

Parameters	Characteristic	MS	Controls	*p* Values
		Mean (SD)	Mean (SD)	
Demographics				
Age (in years)		54.07 (8.46)	50.71 (12.08)	0.585
Sex	Females/Males	10/4	4/3	0.513
Body mass index (in kg^−1^·m^−1^)		27.74 (7.56)	27.44 (3.76)	1.0
Years since MS diagnosis		16.57 (9.69)	NA	NA
Ambulatory assistive device used	Cane	6	0	NA
	2 Canes or Walker	8	0	NA
Type of MS (n)	SPMS	11	NA	NA
	PPMS	3	NA	NA
Biomarkers †				
BDNF (ng·mL^−1^)	At rest	56.56 (25.12)	57.63 (9.48)	0.967
	Post-Pre	−0.09 (18.99)	−0.43 (14.79)	0.837
IL-6 (ng·mL^−1^)	At rest	0.0015 (0.002)	0.0003 (0.0004)	0.010
	Post-Pre	0.0007 (0.0007)	0.0005 (0.0007)	0.384
BDNF/IL-6 ratio	At rest	119808.50 (116307.59)	393501.90 (188730.57)	0.004
	Post-Pre	−70038.97 (95439.75)	−7787.75 (278552.43)	1.0
Functional measures				
Comfortable walking speed (m·s^−1^) ‡		0.32 (0.13)	0.96 (0.26)	<0.001
SF-36 (Vitality/Energy/Fatigue)		37.14 (18.16)	87.14 (9.06)	<0.001
Maximal $\overset{\cdot }{V}O_{2}$ (mL·min^−1^·kg^−1^)		16.35 (6.39)	33.04 (8.95)	<0.001
Duration of GXT (s)		793.29 (259.84)	1087.71 (207.95)	0.046
Maximal workload (Watts)		99.69 (33.84)	271.43 (127.46)	<0.001
Maximal heart rate (beats·min^−1^)		131.57 (23.16)	168.26 (16.83)	0.002

kg: kilogram; m: meter; NA: not applicable; SPMS: secondary-progressive MS; PPMS: primary-progressive MS; BDNF: brain-derived neurotrophic factor; ng: nanogram; mL: milli-liter; IL-6: interleukin-6; s: second; SF: short form; $\overset{\cdot }{V}O_{2}$: oxygen consumption; min: minute; GXT: graded exercise test; † Unable to draw blood samples from two MS participants; ‡ One MS participant was not able to walk over ground more than few steps, and hence, we were unable to measure walking speed.

## Data Availability

Data are available on reasonable request from the corresponding author at the Memorial University of Newfoundland, Canada.

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
