# Peer review of "Fitness Shifts the Balance of BDNF and IL-6 from Inflammation to Repair among People with Progressive Multiple Sclerosis"

_biomolecules, 2021, doi:10.3390/biom11040504_

Round 1

Reviewer 1 Report

The manuscript by  Devasahayam et al. describes the effect of GXT on serum levels of BDNF and IL-6 as well as various parameters of fittness in progressive MS patients and matched control subjects.

As there is no good treatment option for progressive MS patients thus far, improving their fitness is certainly helpful to guarantee quality of life. In this study the authors try to link the effect of GXT to certain serum proteins as a possible mechanism of action.

Issues:

  1. the number of control subjects is too limited. By visual inspection of the data, in particular Fig 1a,b. it is clear that there is no much difference in effect of GXT on the blood measures between MS patients and control subjects. The low n of the control subjects prohibits significance. As the authors claim that the power analysis requests a n of 13-37 subjects, the number of control subjects should be increased.
  2. The interpretation of Fig. 2 is misleading as it is clear that the datapoints of the control subjects and MS patients are separated, one cannot draw a general conclusion. The data should be split between controls and MS.
  3. Discussion: paragraph 4.1 tries to explain why MS patients after GXT did not increase their BDNF levels. The final sentence may be key, also control subjects did not increase BDNF levels, which may indicate that their protocol was not in place to find clear effects on either serum parameters and on fitness read outs.
  4. paragraph 4.3: the interpretation of serum IL-6 levels that were higher in MS patients than in controls  at rest and further increased after GXT, should be better supported than just literature. If the increase in IL-6 after GXT is beneficial, they should measure TNFa in the serum as well and see if that has decreased as suggested in this paragraph. 

Reviewer 2 Report

This is an outstanding paper. It truly adds to the understanding of fitness, exercise, and potential neuroprotection and repair mechanisms in MS. The scientific rigor which Dr Ploughman approaches these questions is much needed and is very clear in this manuscript.

My only recommendation is that the language around fatigue and vitality be made more clear. There are inconsistencies in how the measurement of fatigue/vitality is described. The scale used measures vitality/fatigue on a continuum with higher scores indicating greater vitality, lower fatigue. However, in the abstract it describes higher scores as higher fatigue which is misleading. Because readers unfamiliar with the scale, or those of us who think of higher vitality as better, and lower fatigue as better (no matter how a scale might measure it), I strongly suggest they use the word vitality when referring to scores because that is consistent with higher being better in everyday thinking. As it reads now it sounds like fatigue worsened whereas higher 'fatigue' means higher scores on the vitality/fatigue scale.

Round 2

Reviewer 1 Report

The revised version of the manuscript of Devasahayam et al. shows some textual improvements. As there are quite some limitations to the study design, I would clearly recommend to change the framing of the study. There is no info that supports a transition from an inflammatory to repair status in the patients, except that the BDNF/IL-6 ratio is altered after  exercise. The lack of effect on TNFa levels indicate the absence of such transition, even though it is just one parameter, but that is what the authors focus on in the discussion. Thus, change of title and discussion/conclusion is essential.